# Assessment of the association between body composition and risk of non-alcoholic fatty liver

Mohammad Ariya [1,2], Farbod Koohpayeh[3], Alireza Ghaemi[4], Saeed Osati[5], Sayed Hossain Davoodi[5], Jalaledin Mirzay Razzaz[5], Gholamali Javedan[6,7], Elham Ehrampoush[1,2], Reza Homayounfar[1,2,5]*

1 Noncommunicable Diseases Research Center, Fasa University of Medical Sciences, Fasa, Iran, 2 Department of Nutrition, Fasa University of Medical Sciences, Fasa, Iran, 3 Students Research Committee, Fasa University of Medical Sciences, Fasa, Iran, 4 Faculty of Public Health, Department of Basic Sciences and Nutrition, Health Sciences Research Center, Addiction Institute, Mazandaran University of Medical Sciences, Sari, Iran, 5 Faculty of Nutrition Sciences and Food Technology, National Nutrition and Food Technology Research Institute, Shahid Beheshti University of Medical Sciences, Tehran, Iran, 6 Food Health Research Center, Hormozgan University of Medical Sciences, Bandar 'Abbas, Iran, 7 Minimally Invasive Surgery Research Center, Iran University of Medical Sciences, Tehran, Iran

* r_homayounfar@yahoo.com, homayounfar@sbmu.ac.ir

**Data Availability Statement:** The datasets used and/or analyzed during the current study are available as a supporting file.

## Abstract

Non-alcoholic fatty liver disease (NAFLD) is defined as the condition of fat accumulation in the liver. This cross-sectional study aimed to investigate the relationship between body composition and fatty liver and determine of cut-off point for predicting NAFLD. Samples were selected from the nutrition clinic from 2016 to 2017 in Tehran, Iran. The liver steatosis was calculated using the CAP score through the FiroScan™ and body composition was measured using the dual-energy X-ray absorptiometry scan method. A total of 2160 patients participated in this study, 745 (34.5%) subjects had NAFLD. We found that fat-free tissue was inversely and fat tissue was directly correlated with the risk of NAFLD in almost all factors and the risk of developing NAFLD increases if the total fat exceeds 32.23% and 26.73% in women and men and abdominal fat exceeds 21.42% and 13.76% in women and men, respectively. Finally, we realized that the total fat percent had the highest AUC (0.932 for men and 0.917 for women) to predict the risk of NAFLD. Overall, the likelihood of NAFLD development rose significantly with increasing the amount of total fat and abdominal fat from the cut-off point level.

## Introduction

Non-alcoholic fatty liver disease (NAFLD) is defined as the condition of fat accumulation in the liver with no history of extreme alcohol abuse (< 30 g/d in men and < 20 g/d in women) and in the absence of other liver diseases [1,2]. The disease range varies from NAFLD to non-alcoholic steatohepatitis (NASH) [1]. NASH is a severe clinical form of NAFLD that has become one of the major causes of liver transplant [3]. As in other liver diseases, NAFLD can

**Funding:** The study was supported by the Fasa University of Medical Sciences (Grant No.: 97107).

**Competing interests:** The authors hereby affirm that the manuscript is original, that all statements asserted as facts are based on authors' careful investigation and accuracy, that the manuscript has not been published in total or in part previously and has not been submitted or considered for publication in total or in part elsewhere. Each author acknowledges he/she has participated in the work substantively and is prepared to take public responsibility for the work and authors have no competing interest in the results of the article.

**Abbreviations:** ALT, alanine aminotransferase; AST, aspartate aminotransferase; BFP, body fat percentage; BMI, body mass index; DAG, diacylglycerol; dB, decibel; DBP, Diastolic Blood Pressure; DXA, dual-energy X-ray absorptiometry; FBS, fasting blood sugar; FF, Friedewald formula; FFA, free fatty acids; FFM, fat-free mass; FFQ, food frequency questionnaire; FLI, fatty liver index; FM, fat mass; GGT, Gamma-glutamyl transferase; HDL-C, high density lipoprotein cholesterol; KC, Kupffer cells; LSM, Liver stiffness measurement by FibroScan; MET, Metabolic Equivalent; NAFLD, Non-alcoholic fatty liver disease; NASH, non-alcoholic steatohepatitis; NF, nuclear factor; SBP, Systolic Blood Pressure; TC, total cholesterol; TG, triglyceride; VF, visceral fat; WC, waist circumference.

also induce hepatic fibrosis and lead to such diseases as cirrhosis, liver and colorectal cancer, cardiovascular disease, type 2 diabetes, insulin resistance, and obesity, or exacerbate their progression [4–6]. Of these, approximately 10–25% and 5–8% of these patients develop NASH and liver cirrhosis, respectively, within 5 years [2].

On the other hand, the severity of fatty liver in these patients is associated with impaired blood glucose status [7]. NAFLD is also predicted to be a major cause of liver transplantation over the next 10 years [8]. Studies suggest that the prevalence of NAFLD is influenced by age, sex, ethnicity, and geographic area [9]. Besides, various studies have examined the association between obesity and NAFLD. In this regard, a meta-analysis study reported that metabolism-related serological indicators in people with NAFLD who have BMI over 25were significantly higher than those of lean NAFLD patients [10]. Also, another systematic review study demonstrated that people with NAFLD who have BMI over 25 also had a higher risk of liver fibrosis and NASH than lean individuals [11] and that significant weight loss (a 7%-10% reduction in BMI) reduced the progression of hepatic fibrosis [12]. Some studies, on the other hand, do not confirm such a relationship and suggest that obesity does not lead to advanced liver disease and NASH in patients with NAFLD [6].

Although the disease is more prevalent (about 80%) in individuals with obesity and obesity is an important risk factor for its development with greater complications [11], it is also expected to occur in lean and even normal-weight individuals [10]. As such, significant percentages of people with this disease in Asia have a low BMI and in Asians, WHO suggests a lower BMI cut-off as a risk factor for non-communicable diseases[13,14]. Furthermore, the diagnosis may be delayed since the risk factors for fatty liver present in individuals with obesity do not exist in lean individuals [10]. A meta-analysis study in this area showed that the risk of metabolic abnormalities and cardiovascular disease was not in general significantly different between obese and lean subjects with NAFLD [10]. The study eventually suggested that abdominal fat, or central obesity, could be a major factor in the pathogenesis of NAFLD in lean-NAFLD individuals [10]. Central obesity or visceral fat (VF) [determined by waist circumference (WC)] is defined as the presence of excess fat in the abdomen, and this type of obesity is often associated with the development and progression of NAFLD or more advanced forms of liver disease [15]. Thus, measurement of body composition rather than BMI may be helpful in the prediction of NAFLD [5,15].

Nowadays, increasing NAFLD in Iran is considered a public health problem. Therefore, it is expected that this issue is a health priority to be further addressed with more studies on its risk factors, including environmental and nutritional factors. Other studies have either examined BMI or obesity and fatty liver [16] or generally studied body fat [6,17]. Yet, none of them investigated the relationship between different organs of the body composition separately with fatty liver and thereby does not predict a cut-off point for any. The present study aimed to investigate the relationship in different organs between body composition together with fatty liver in patients with NAFLD and the group without it, and determine a cut-off point based on total fat and abdominal fat, in Tehran, Iran.

## Materials and methods

### Study design and sample

The present study was a cross-sectional study in which samples were randomly selected using a convenience sampling method from clients of a university-affiliated nutrition clinic in Tehran-Iran from April 2016 to late September 2017. During this time, 2842 individuals were referred to the clinic for nutritional counseling all of whom were new cases evaluated at primary visits. The present findings are part of a larger study which investigated factors affecting

fatty liver in an Iranian population (Tehran), which is still in progress. This clinic is a private center that offers a wide variety of nutrition therapies such as weight loss, eating disorders and sports nutrition. It is noteworthy that participants were selected through stratified random sampling to include an acceptable proportion of all age and sex groups with Iranian nationality.

The inclusion criteria were consent for participation and no history of extreme alcohol abuse ($< 30$ g/d in men and $< 20$ g/d in women). Exclusion criteria included one of the diseases affecting the patient's metabolic status, such as cancer, liver disease (including liver cancer and hepatitis), diabetes diagnosed definitively, consumption of diabetes medicines, thyroid disorders, and chemotherapy procedures. We also excluded people who were on a diet in the last three months and took blood sugar/fat lowering pills, as well as participants younger than 18 and over 65 years of age.

The study protocol followed the Helsinki Declaration and was confirmed by the Ethics Committee of Fasa University of Medical Sciences (Approval Code: IR.FUMS.REC.1397.172). The participants were informed about the research objectives and the written informed consent was obtained from the subjects before starting the survey.

### Data collection

All participants were initially briefed about the aims of this study and its potential benefits. Then, informed consent was obtained from those who were willing to participate. Afterwards, a demographic questionnaire, including age, sex, smoking status, and menopausal status of women, was completed by a trained questioner. Then, a validated food frequency questionnaire (FFQ), containing 168 food items, was filled out for all participants [18]. In addition to food items, the questionnaire also included portion size information for the measurements of standard food size. The questionnaire was completed based on the frequency of food items used during the past day, month and year as well as on the standard portion size. We converted the values collected for each meal to grams per day (g/d) using a household scale manual and calculated energy consumption by the subjects. To calculate the activity level of individuals, each activity was multiplied by the number of implementation days and duration of the activity, which is reported by the Metabolic Equivalent (MET) unit.

Anthropometric indices (height, weight, BMI and WC) were measured by a trained expert using standard instruments including a tape meter, stadiometer (0.1 cm precision), and a 767-digital scale ($\pm$ 0.1 kg; Seca, Japan). BMI was obtained by dividing weight (kg) by the square of the height ($m^2$). WC was taken at the midway between the superior border of the iliac crest and the lowest rib in a standing position and after exhaling. Thereafter, blood pressure was measured after a 15-min rest in a sitting position (sphygmomanometer, mercury, ALPK1, Japan). For accuracy, this assessment was repeated after 15 minutes and the mean of both measurements was recorded as the blood pressure of each subject.

Subsequently, participants were referred to a central laboratory to perform relevant tests [fasting blood sugar (FBS), triglyceride (TG), total cholesterol (TC), and high-density lipoprotein cholesterol (HDL-C), Gamma-glutamyl transferase (GGT), aspartate aminotransferase (AST) and alanine aminotransferase (ALT)]. For these, 10 ml of venous blood was taken after 10 h of fasting, and all tests were performed by Pars Azmoon kits-Iran. GGT was measured by enzymatic methods base on SZASZ's approach, and AST and ALT by the kinetic method. HDL-C, TG, and total cholesterol levels were also determined by the enzymatic method, and then reported in mg/dl. Besides, the Friedewald formula (FF) was employed to calculate low-density lipoprotein cholesterol (LDL-C) [19].

The liver steatosis was calculated using FiroScan™ (Echosens, Paris, France) by a reliable sonographer, XL probe in this case [20]. Besides, for measuring fat in the liver, the Controlled Attenuation Parameter (CAP) with the decibels per meter (dB/m) units was used (Pearson correlation = 0.698, P-value < 0.001). According to the results of CAP scores, participants were divide into four groups, where normal liver or S0 steatosis grade was defined at CAP< 238 dB/m which indicates 0 to 11% of the fat in the liver. Mild (S1 steatosis grade) and moderate fatty liver (S2 steatosis grade) are determined at 238–260 dB/m and 260–293 dB/m which shows 11 to 33% and 33 to 66% of fat in the liver, respectively. In this regard, severe fatty liver or S3 steatosis grade occurs at CAP> 293 dB/m, indicating above 66% of the fat exist in the liver. Liver stiffness measurement by FiroScan™ (LSM) was reported in decibel (dB).

### Body composition measurement

There are different methods to measure body composition, among which the dual-energy X-ray absorptiometry (DXA) technique has been recommended to be suitable for measuring body composition, including fat mass (FM) and fat-free mass (FFM) [21]. On the other hand, another study has suggested the DXA method for assessing body composition and VF distribution in obese patients with NAFLD [22]. DXA estimates body composition by the absorption of photons in tissues and reports total body weight in FM, FFM and bone minerals [22].

In the present study, body composition was measured through a Lunar instrument (Lunar iDXA, GE Healthcare, Wisconsin, USA), which evaluates the whole body using DXA scanning, and is a standard tool for body mass measurement [22,23]. Fat mass and fat-free mass in the right and left arms, left and right thighs, trunk, and abdomen were measured by this device; also, it measured total fat and total fat-free mass, all of which were reported in grams. It should be noted that the subjects were scanned by this device in the supine position for 15 minutes with no additional movement according to the manufacturer's instructions.

$$\mathrm{DEXA\ fat-free\ mass\ (DEXA\_FFM)} = \mathrm{lean\ mass} + \mathrm{BMC\ (bone\ mineral\ content)}.$$

$$\mathrm{DEXA\ fat\ mass\ (DEXA\_FM)} = \mathrm{fat\ mass}.$$

### Statistical analyses

Based on the results of normal data distribution, mean values (± SD) were used for quantitative variables. Univariate variables were analyzed by independent samples t-tests. The relationship between fatty liver with body composition was examined by the Pearson correlation test. Curve analysis was also performed using receiver operating characteristic (ROC) to obtain cut-off points and area under the ROC curve (AUC) for associations between abdominal and total fat with NAFLD. Data analysis was performed in SPSS version 16 and P-value of less than 0.05 was considered significant.

### Results

A total of 2842 individuals were referred to the clinic, among whom 2160 patients participated in this study and 682 individuals did not intend to participate or had an exclusion criterion. Among all participants, 745 (34.5%) subjects had NAFLD, while 1415 (65.5%) did not. Table 1 (S1 and S2 Tables by sex) shows demographic data, results of laboratory tests, and

**Table 1. Demographic and basic information of the participants in the group without NAFLD and NAFLD patients.**

| | The group without NAFLD (N = 1415) | | NAFLD patient (N = 745) | | P-value* | OR | 95% C.I. | |
|---|---|---|---|---|---|---|---|---|
| | **Mean** | **SD** | **Mean** | **SD** | | | **Lower** | **Upper** |
| Age (years) | 35 | 9 | 49 | 8 | <0.001 | 1.19 | 1.173 | 1.207 |
| Weight (Kg) | 80.8 | 12.2 | 96.1 | 11.5 | <0.001 | 1.103 | 1.093 | 1.113 |
| BMI (Kg/m$^2$) | 25.4 | 4.1 | 30.9 | 3.4 | <0.001 | 1.413 | 1.371 | 1.457 |
| WC (cm) | 96.49 | 16.04 | 118.10 | 12.77 | <0.001 | 1.097 | 1.088 | 1.106 |
| Physical Activity (MET/24h) | 24.9 | 5.0 | 18.3 | 4.5 | <0.001 | 0.762 | 0.744 | 0.781 |
| Energy (Kcal) | 2405.49 | 413.73 | 2664.98 | 460.11 | <0.001 | 1.001 | 1.001 | 1.002 |
| FBS (mg/dL) | 99.39 | 13.53 | 118.29 | 13.15 | <0.001 | 1.102 | 1.092 | 1.111 |
| LDL-C (mg/dL) | 93.72 | 15.56 | 113.71 | 16.61 | <0.001 | 1.077 | 1.069 | 1.085 |
| HDL-C (mg/dL) | 48.18 | 7.95 | 38.88 | 7.08 | <0.001 | 0.858 | 0.846 | 0.871 |
| TG (mg/dL) | 186.57 | 26.15 | 214.28 | 25.03 | <0.001 | 1.041 | 1.037 | 1.045 |
| TC (mg/dL) | 176.01 | 17.11 | 195.40 | 16.18 | <0.001 | 1.072 | 1.064 | 1.079 |
| SBP (mmHg) | 120.47 | 10.63 | 130.09 | 10.66 | <0.001 | 1.256 | 1.188 | 1.327 |
| DBP (mmHg) | 80.04 | 10.02 | 80.84 | 9.6 | <0.001 | 2.205 | 1.997 | 2.435 |
| GGT (mg/dL) | 24.17 | 10.60 | 35.87 | 13.24 | <0.001 | 1.085 | 1.075 | 1.094 |
| ALT (UL/L) | 40.55 | 14.55 | 57.92 | 13.71 | <0.001 | 1.083 | 1.075 | 1.092 |
| AST (UL/L) | 32.51 | 14.03 | 49.33 | 12.52 | <0.001 | 1.091 | 1.082 | 1.100 |
| LSM (dB) | 216.08 | 43.29 | 273.86 | 34.84 | <0.001 | - | - | - |
| Left Arm Fat (kg) | 2.577 | 1.63 | 4.066 | 1.55 | <0.001 | 1.703 | 1.604 | 1.808 |
| Right Arm Fat (kg) | 2.646 | 1.69 | 4.201 | 1.59 | <0.001 | 1.685 | 1.590 | 1.786 |
| Left Leg Fat (kg) | 3.802 | 2.57 | 6.644 | 2.37 | <0.001 | 1.514 | 1453 | 1.577 |
| Right Leg Fat (kg) | 3.820 | 2.61 | 6.679 | 2.38 | <0.001 | 1.503 | 1.443 | 1.564 |
| Abdominal Fat (kg) | 6.456 | 4.18 | 12.072 | 3.47 | <0.001 | 5.646 | 4.535 | 7.029 |
| Total Fat (kg) | 22.309 | 9.62 | 37.018 | 7.84 | <0.001 | 1.216 | 1.197 | 1.236 |
| Left Arm FatFree (kg) | 1.851 | .46 | 1.809 | .42 | .039 | 0.813 | 0.668 | 0.990 |
| Right Arm FatFree (kg) | 1.862 | .46 | 1.819 | .42 | .036 | 0.808 | 0.662 | 0.986 |
| Left Leg FatFree (kg) | 4.563 | .62 | 4.506 | .66 | .048 | 0.870 | 0.757 | 0.999 |
| Right Leg FatFree (kg) | 4.597 | .63 | 4.504 | .62 | .001 | 0.789 | 0.685 | 0.910 |
| Abdominal FatFree (kg) | 11.686 | 1.33 | 11.100 | 1.12 | <0.001 | 0.692 | 0.643 | 0.745 |
| Total FatFree (kg) | 23.994 | 2.99 | 23.260 | 2.49 | <0.001 | 0.910 | 0.881 | 0.940 |
| | **N** | **%** | **N** | **%** | | | | |
| Sex | Female | 656 | 62.1% | 401 | 37.9% | 0.001 | 0.741 | 0.621 | 0.886 |
| Smoking | Yes | 240 | 17% | 125 | 16.8% | 0.914 | 1.013 | 0.799 | 1.284 |

BMI, body mass index; FBS, Fasting Blood Sugar; HDL-C, high-density lipoprotein cholesterol; LDL-C, low-density lipoprotein cholesterol; TG, Triglyceride; TC, Total cholesterol; SBP, Systolic Blood Pressure; DBP, Diastolic Blood Pressure; GGT, gamma-glutamyl transferase; ALT, alanine aminotransferase; AST, aspartate aminotransferase; LSM, liver stiffness measurement by FibroScan.

*. Independent samples t-tests.

anthropometric characteristics of the study population. The table also reveals the result of liver stiffness measurement by FibroScan™ (LSM). Subjects with NAFLD had significantly lower physical activity and HDL-C, but higher LDL-C, TG, TC, blood glucose, blood pressure, and energy intake than those without NAFLD. Diastolic hypertension had the highest association with NAFLD. Besides the factors mentioned in Table 1, fat-free mass and fat mass were scanned through DXA [based on kilogram (kg)] in the different organs (thigh, arm, and abdomen), and OR (odds ratio) were calculated.

**Table 2. Correlation# of fatty liver by body composition in women.**

| | NAFLD | Left Arm Fat | Right Arm Fat | Left Leg Fat | Right Leg Fat | Trunk Fat | Total Fat | Left Arm FatFree | Right Arm FatFree | Left Leg FatFree | Right Leg FatFree | Trunk_FatFree | Total FatFree |
|---|---|---|---|---|---|---|---|---|---|---|---|---|---|
| **NAFLD** | 1 | .733** | .742** | .748** | .749** | .756** | .750** | -.167** | -.167** | -.076* | -.130** | -.326** | -.289** |
| **Left Arm Fat** | .733** | 1 | .937** | .934** | .936** | .952** | .940** | -.243** | -.245** | -.130** | -.158** | -.453** | -.353** |
| **Right Arm Fat** | .742** | .937** | 1 | .943** | .949** | .957** | .949** | -.250** | -.250** | -.142** | -.179** | -.462** | -.380** |
| **Left Leg Fat** | .748** | .934** | .943** | 1 | .948** | .956** | .949** | -.241** | -.262** | -.132** | -.172** | -.451** | -.373** |
| **Right Leg Fat** | .749** | .936** | .949** | .948** | 1 | .956** | .953** | -.242** | -.255** | -.133** | -.182** | -.457** | -.375** |
| **Trunk Fat** | .756** | .952** | .957** | .956** | .956** | 1 | .966** | -.242** | -.253** | -.130** | -.172** | -.464** | -.380** |
| **Total Fat** | .750** | .940** | .949** | .949** | .953** | .966** | 1 | -.246** | -.239** | -.125** | -.179** | -.447** | -.366** |
| **Left Arm FatFree** | -.167** | -.243** | -.250** | -.241** | -.242** | -.242** | -.246** | 1 | .335** | .205** | .276** | .375** | .347** |
| **Right Arm FatFree** | -.167** | -.245** | -.250** | -.262** | -.255** | -.253** | -.239** | .335** | 1 | .284** | .302** | .376** | .366** |
| **Left Leg FatFree** | -.076* | -.130** | -.142** | -.132** | -.133** | -.130** | -.125** | .205** | .284** | 1 | .260** | .290** | .257** |
| **Right Leg FatFree** | -.130** | -.158** | -.179** | -.172** | -.182** | -.172** | -.179** | .276** | .302** | .260** | 1 | .312** | .252** |
| **Trunk_FatFree** | -.326** | -.453** | -.462** | -.451** | -.457** | -.464** | -.447** | .375** | .376** | .290** | .312** | 1 | .452** |
| **Total FatFree** | -.289** | -.353** | -.380** | -.373** | -.375** | -.380** | -.366** | .347** | .366** | .257** | 252** | .452** | 1 |

**. Correlation is significant at the 0.01 level (2-tailed);

*. Correlation is significant at the 0.05 level (2-tailed);

#. Pearson correlation test. NAFLD; Non-alcoholic fatty liver disease.

The relationship between NAFLD with each factor of body composition was examined separately in women and men by the Pearson correlation test (Tables 2 and 3). It was found that fat-free tissue was inversely correlated with the risk of NAFLD in all factors of both women and men, except the right arm fat-free in men. The highest correlation was observed with trunk fat-free and total fat-free in both males and females, respectively. Tables 2 and 3 also show that the likelihood of NAFLD increases with thicker fat tissue in all factors, and the uppermost relationships were observed between trunk fat and total fat with NAFLD in both men and women.

Table 4 demonstrates the cut-off point for predicting NAFLD in men and women based on the percentage of total fat and abdominal fat. As shown in the table, the risk of developing NAFLD increases if the total fat exceeds 32.23% in women and 26.73% in men. Moreover, the risk of developing NAFLD rises if the abdominal fat exceeds 21.42% and 13.76% in females and males, respectively.

Fig 1 reveals ROC curves for NAFLD using body fat percent in women and men. The ROC curve was calculated to compare the predictive power of body fat percentage (BFP) for the risk of NAFLD. The cut-off points of body fat percent showing maximum sensitivity for detecting the NAFLD was total fat present in men (sensitivity 91.28%, specificity 77.60%) and then in women (sensitivity 90.02%, Specificity 81.10%). Besides, among four factors of BFP, total fat percent appeared to have the highest AUC (0.932 for men and 0.917 for women) to predict the risk of NAFLD.

## Discussion

To our knowledge, this is one of the first studies to compare NAFLD with body composition in an Iranian population. As a novelty, we compared different organs of fat and fat-free tissues

**Table 3. Correlation# of fatty liver by body composition in men.**

| | NAFLD | Left Arm Fat | Right Arm Fat | Left Leg Fat | Right Leg Fat | Trunk Fat | Total Fat | Left Arm FatFree | Right Arm FatFree | Left Leg FatFree | Right Leg FatFree | Trunk_FatFree | Total FatFree |
|---|---|---|---|---|---|---|---|---|---|---|---|---|---|
| **NAFLD** | 1 | .745** | .745** | .744** | .751** | .769** | .757** | -.025 | .017 | -.022 | -.103** | -.189** | -.110** |
| **Left Arm Fat** | .745** | 1 | .837** | .854** | .853** | .878** | .872** | -.086** | -.056 | -.068* | -.104** | -.264** | -.160** |
| **Right Arm Fat** | .745** | .837** | 1 | .863** | .849** | .889** | .872** | -.114** | -.070* | -.080** | -.126** | -.282** | -.186** |
| **Left Leg Fat** | .744** | .854** | .863** | 1 | .881** | .902** | .887** | -.112** | -.056 | -.074* | -.122** | -.280** | -.187** |
| **Right Leg Fat** | .751** | .853** | .849** | .881** | 1 | .903** | .871** | -.080** | -.051 | -.064* | -.128** | -.288** | -.179** |
| **Trunk Fat** | .769** | .878** | .889** | .902** | .903** | 1 | .905** | -.114** | -.078** | -.080** | -.145** | -.311** | -.200** |
| **Total Fat** | .757** | .872** | .872** | .887** | .871** | .905** | 1 | -.101** | -.061* | -.072* | -.137** | -.282** | -.179** |
| **Left Arm FatFree** | -.025 | -.086** | -.114** | -.112** | -.080** | -.114** | -.101** | 1 | .232** | .251** | .207** | .320** | .292** |
| **Right Arm FatFree** | .017 | -.056 | -.070* | -.056 | -.051 | -.078** | -.061* | .232** | 1 | .227** | .237** | .370** | .316** |
| **Left Leg FatFree** | -.022 | -.068* | -.080** | -.074* | -.064* | -.080** | -.072* | .251** | .227** | 1 | .248** | .257** | .303** |
| **Right Leg FatFree** | -.103** | -.104** | -.126** | -.122** | -.128** | -.145** | -.137** | .207** | .237** | .248** | 1 | .261** | .261** |
| **Trunk_FatFree** | -.189** | -.264** | -.282** | -.280** | -.288** | -.311** | -.282** | .320** | .370** | .257** | .261** | 1 | .436** |
| **Total FatFree** | -.110** | -.160** | -.186** | -.187** | -.179** | -.200** | -.179** | .292** | .316** | .303** | .261** | .436** | 1 |

**. Correlation is significant at the 0.01 level (2-tailed);

*. Correlation is significant at the 0.05 level (2-tailed);

#. Pearson correlation test. NAFLD; Non-alcoholic fatty liver disease.

separately with the risk of NAFLD and determined the cut-off point based on total and abdominal fat percentage. The present study revealed that increased adipose tissue and decreased fat-free tissue in almost all organs increased the likelihood of NAFLD. Finally, our observations indicated that the risk of NAFLD development rises if total fat increases by 32.23% in women and 26.73% in men, which had maximum sensitivity and predictability for detecting the NAFLD.

The main difference between the present and previous studies [16] is the use of body composition rather than BMI. However, one study has even reported a significant linear relationship between NAFLD and BMI (above 25 kg/m$^2$) [16]. Even so, being lean or having a BMI less than 18.5 kg/m$^2$ does not necessarily mean being healthy lean because a large amount of abdominal adipose tissue was reported in lean-NAFLD compared to lean NAFLD-free subjects [10]. On

**Table 4. Receiver operating characteristic curve (ROC) analysis for defining the ideal fat percentage of body weight cut-off point based on nonalcoholic fatty liver disease.**

| | | Cut-off Point | Sensitivity | Specificity | AUC | PPV | NPV | PLR | NLR |
|---|---|---|---|---|---|---|---|---|---|
| Female | **Total fat** | **32.23** | 90.02 | 81.10 | 0.932 | 74.4 | 93.0 | 4.76 | 0.12 |
| | **Abdominal fat** | **21.42** | 76.81 | 85.06 | 0.878 | 75.9 | 85.7 | 5.14 | 0.27 |
| Male | **Total fat** | **26.73** | 91.28 | 77.60 | 0.917 | 65.1 | 95.0 | 4.11 | 0.12 |
| | **Abdominal fat** | **13.76** | 90.70 | 76.94 | 0.898 | 64.1 | 94.8 | 3.93 | 0.12 |

AUC: Area under the curve; PPV: Positive predictive value; NPV: Negative predictive value; PLR: Positive likelihood ratio; NLR: Negative likelihood ratio.

The optimum cut-off was defined as the point on the ROC curve with the maximum sensitivity + specificity (Youden method).

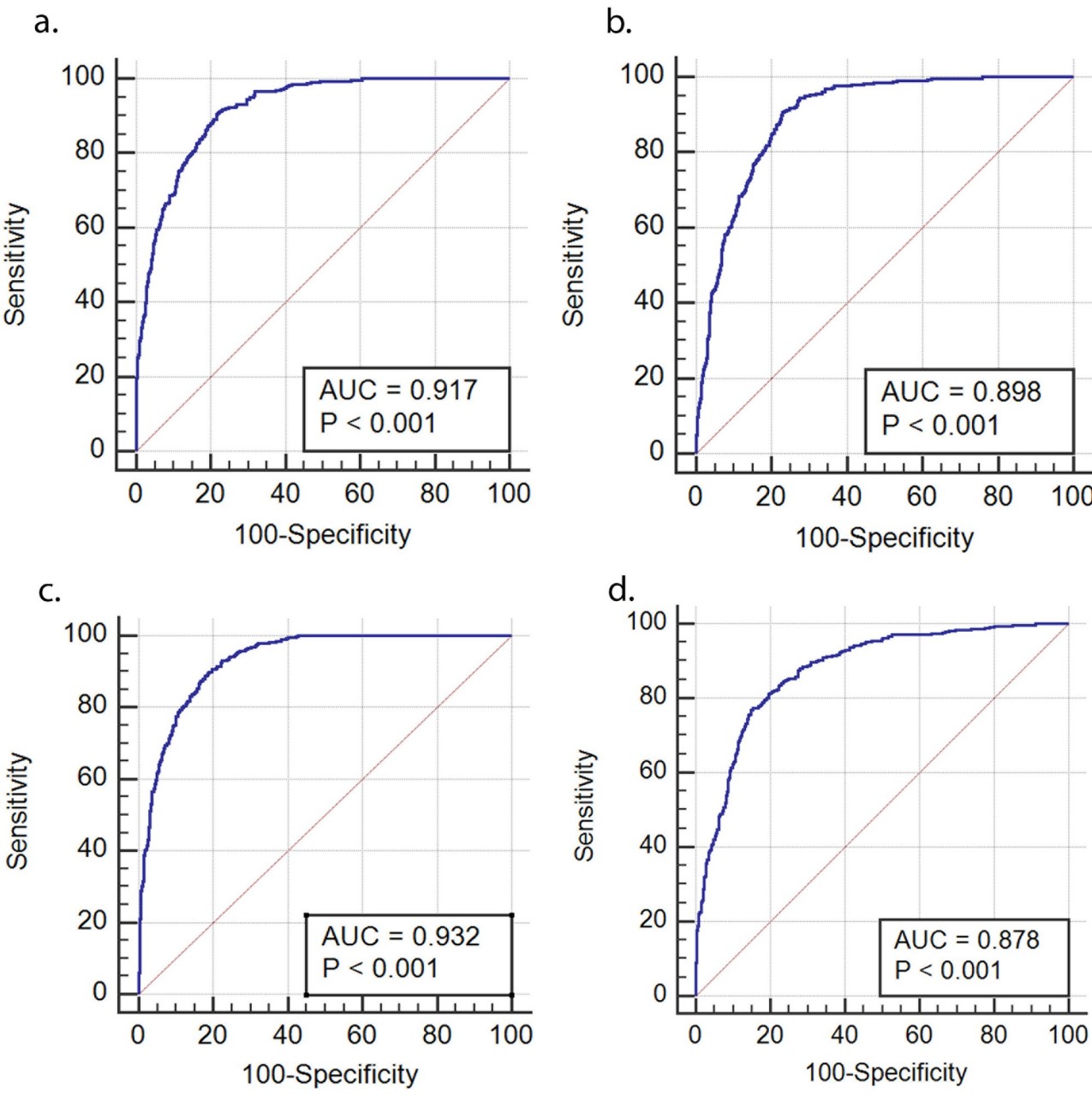

**Fig 1. ROC curves for NAFLD using body fat percent.** (a) ROC for Total Fat in Males (b) ROC for Abdominal Fat in Males, (c) ROC for Total Fat in Females (d) ROC for Abdominal Fat in Females.

the other hand, abdominal fat itself appears to be a key factor in determining TG and TC levels in NAFLD, which is nearly identical in both obese and lean individuals with NAFLD [10].

Saida et al. studied 1,851 women and 1,259 men in Japan and found that in addition to BMI, increased BFP was also directly associated with fatty liver disease, and body fat levels were not necessarily high in people with high BMI's [17]. Accordingly, it can be concluded that people who are classified in normal and even lean classes in terms of BMI do not necessarily have low total fat or are less likely to develop NAFLD.

In the present study, there was a significant likelihood of the development of NAFLD if the total fat and abdominal fat, respectively, exceeded 32.23% and 21.42% in women and 26.73% and 13.76% in men. The results of other studies are contradictory. Saida et al. found a significant relationship between BMI and BFP with fatty liver in men, but this association was not significant in women (despite a higher BFP than that of men) [17]. On the other hand, other studies reported noteworthy results indicating that although obesity increased in men with age, BFP rose only marginally. It was opposite in women, i.e., reduced obesity but increased mean BFP with age [24,25]. In another study, Motamed et al. reported that the prevalence of NAFLD was higher in men under 40 years of age than in women, but it was more prevalent in women over 60 years of age [26]. These differences may be explained by the lower muscle mass in women than men [17], differences in sex hormones [26], and different body composition, hence the incidence of NAFLD varies between women and men.

Notably, percentage of total fat in women and men of the same age is remarkably different. In fact, total fat is divided into "essential" and "storage" fats. Essential fat has been defined as the fundamental fat for normal physiologic functioning which consists of around 3% and 12% total body of males and females, respectively. On the other hand, storage fat is considered an energy reserve which is expendable and stored in adipose tissue. Thus, the average total fat of an individual (essential fat plus storage fat) in men and women is estimated to be 18–24 percent and 25–31 percent, respectively [27]. Since the percentage of total fat in women is higher than men due to their physiological condition, the amount of total fat affected in diseases such as NAFLD should typically be higher in them than in males. Similarly, in men, the lower amount of total and abdominal fat affects the development of fatty liver, which does not necessarily mean that they develop NAFLD easily.

According to our findings, there was an increased risk of NAFLD if the percentage of abdominal or VF and total fat exceeded a certain cut-off point, which was not different in obese or lean individuals. Matsuzawa et al. stated that as VF has a higher metabolism than subcutaneous fat, it mobilizes free fatty acids (FFA) to the liver [28], which is an important risk factor for NAFLD development. Similarly, Sogabe et al. [29], Abenavoli et al. [15], and Bouchi et al. [30] reported that VF or central obesity is a major factor in the development of NAFLD. Besides, other investigations have identified VF or abdominal fat as a key factor in the incidence of NAFLD or other liver diseases [31,32]. In this regard, Ko et al. found that increases in BFP and VF raised the risk of NAFLD by 6.5 and 17.8 times, respectively [32]. However, some studies reported contrasting results. Choudhary et al., for example, concluded that both total adipose tissue and subcutaneous adipose tissue volume were correlated with the severity of NAFLD, but no relationship was found between VF volume and NAFLD [33]. It seems that more studies are needed in this respect.

Based on our data, fat-free tissue had a preventive effect on different organs and, conversely, adipose tissue in different tissues increased the likelihood of NAFLD. Nonetheless, the highest risks were recorded in trunk fat and total fat, and the protective properties were uppermost in trunk fat-free and total fat-free, respectively, in both men and women. In another study based on DXA analysis, abdominal circumference and waist-to-height ratio were associated with trunk fat mass in obese NAFLD patients [34], which is consistent with the present study. Likewise, another study reported that, although body fat and trunk fat percentages were significantly different between subjects with and without NAFLD, there were no significant differences in the extremity fat percentage between the two groups [22]. Differences in studied populations and the type of device used to measure body composition might have caused these differences.

In the current trial, the levels of TG, LDL-C, and TC were significantly higher in the NAFLD group, and HDL-C levels were significantly lower than in the group without this

disorder, which differs from those of other reports. A systematic review reported that blood cholesterol was not significantly different between obese and non-obese subjects with NAFLD [6]. Ko et al., on the other hand, observed that elevated TG and lower HDL-C could be considered as a risk factor for NAFLD [32], which is confirmed by the present study. It seems that more studies are needed in this regard.

In the present study, WC, FBS, and blood pressure were significantly higher in subjects with NAFLD. Studies indicate that since BMI does not represent body composition, measuring WC for prediction of NAFLD provides a better insight than using BMI [35]. Ko et al. reported that WC greater than 90 cm in men and above 80 cm in women increased the risk of NAFLD by 9.8 times [32]. Another study reported waist cut-off points of 89 cm in men and 84 cm in women for the risk of NAFLD [34]. Also, Motamed et al. reported that WC measurement is a more appropriate and convenient index for prediction of NAFLD [26]. Other studies also reported results consistent with our research [36].

Based on our result, the percentages of total fat had the highest sensitivity to predict the risk of NAFLD. In other populations, the result was contradicted. For instance, the study conducted in China (n = 3589) showed an independent positive relationship of android fat mass and an inverse relationship of muscle mass with NAFLD [37]. Besides, one research in the Netherlands (n = 4609) reported that both increases in fat mass and decreases in skeletal muscle mass were associated with NAFLD in normal-weight individuals, and fat distribution could be the best predictor for developing NAFLD [38]. The results of both of the two studies mentioned above were consistent with our results. On the other hand, the study carried out in Korea (n = 452) indicated that people with lower muscle mass had a higher risk for developing NAFLD [39] that was inconsistent with the current research. Because of different populations, we cannot generalize findings to other contexts.

Multiple mechanisms have been proposed for increased risk of NAFLD development as a result of abdominal fat or VF. One mechanism explains that the fatty tissue itself releases a large amount of FFA into the bloodstream. Related research presented evidence of FFA contribution from 5 to 10% released from VF in people with normal BMI, which increased to 30% in people with higher VF [40]. This high amount of FFA in liver cells can cause intracellular inflammation, insulin resistance, and hepatocyte steatosis elevation in the liver [41,42]. Furthermore, FFA released from VF can induce inflammation in hepatocytes by stimulating the nuclear factor (NF)-kB pathway and activating Kupffer cells (KC) [43], which can contribute to liver disease progression through raising diacylglycerol (DAG) [44]. Therefore, weight loss [45] or more specifically reduction of VF has been suggested as the first line of treatment for NAFLD, and such patients can be greatly improved by reducing VF [46].

Besides, VF is a source of several cytokines secreted from an adipose tissue called adipokines [47], which plays an important role in the pathogenesis of NAFLD [5]. One of the most important adipokines is the leptin hormone, which is impeded by leptin resistance in these conditions despite its elevated levels both in obesity and in NAFLD and NASH [48]. A major function of leptin is to regulate consumed energy and received energy, and prevent the accumulation of fats in non-adipose tissues such as the liver, and in hepatic insulin resistance indirectly plays a role in preventing liver fibrosis [49,50]. Other adipokines such as adiponectin [51], resistin [52], and visfatin [53] are also involved in NAFLD.

In addition to the aforementioned, another mechanism suggests that hypertrophic adipocytes induce the release of chemokines in VF. These substances, in turn, lead to the accumulation of lipids in VF and ultimately induce low-grade inflammation in this tissue, which can also be an independent factor in the prediction of NAFLD [29,31].

## Limitations

There were some limitations to the present study. First, due to a cross-sectional study, causal relationships could not be derived. Second, since body composition varies across geographic regions, it may not be possible to generalize it to other populations. The present study did not investigate the association between food and beverage consumption with fatty liver development, which may be an important factor in future studies as dietary habits vary in populations. Besides, measuring inflammatory cytokines or blood cell contents could help strengthen the study, but due to financial constraints, we did not examine these items, which requires further studies for more precise results.

## Strengths

The main strength of this study is its relatively larger population size. Besides, a considerable number of NAFLD-related variables enabled us to more closely scrutinize our main variables. Finally, the application of the FibroScan™ device as an established method for the study of NAFLD [20] and the use of the DXA scan as a standard method for estimating body composition [22,23] can further validate our study.

## Conclusion

In summary, our research investigates the relationship between NAFLD with body composition in an Iranian population. It should be noted that based on a literature review, this study is one of the few studies in which both adipose and non-adipose tissues of different organs were separately compared with the fatty liver. According to our observations, there is an elevated risk of NAFLD development if the total fat in men and women exceeds 26 and 32 percent, respectively, which had maximum sensitivity for detecting the NAFLD. Another finding was that the likelihood of NAFLD development rose significantly with thicker abdominal fat by more than 21% in women and 13% in men. As a result, health planners and policymakers can consider this to prevent the development and progression of fatty liver in society. Finally, a longitudinal study is needed to test the ability of abdominal and total fat to predict NAFLD.

## Supporting information

**S1 Table. Demographic and basic information of the male participants in the group without NAFLD and NAFLD patients.**
(DOCX)

**S2 Table. Demographic and basic information of the female participants in the group without NAFLD and NAFLD patients.**
(DOCX)

**S1 Dataset.**
(CSV)

## Acknowledgments

The authors appreciate all people that patiently contributed to this study.

## Author Contributions

**Data curation:** Farbod Koohpayeh, Elham Ehrampoush.

**Funding acquisition:** Alireza Ghaemi.

**Investigation:** Alireza Ghaemi, Sayed Hossain Davoodi, Reza Homayounfar.

**Methodology:** Saeed Osati, Gholamali Javedan.

**Project administration:** Gholamali Javedan, Reza Homayounfar.

**Resources:** Saeed Osati.

**Software:** Reza Homayounfar.

**Supervision:** Jalaledin Mirzay Razzaz.

**Validation:** Jalaledin Mirzay Razzaz, Gholamali Javedan, Reza Homayounfar.

**Visualization:** Sayed Hossain Davoodi, Reza Homayounfar.

**Writing – original draft:** Mohammad Ariya.

**Writing – review & editing:** Mohammad Ariya.

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
