## [Decision Letter · Decision Letter 0]

26 Nov 2020

PONE-D-20-34964

Assessment of the association between body composition and risk of Non-alcoholic fatty liver

PLOS ONE

Dear Dr. Reza Homayounfar,

Thank you for submitting your manuscript to PLOS ONE. After careful consideration, we feel that it has merit but does not fully meet PLOS ONE’s publication criteria as it currently stands. Therefore, we invite you to submit a revised version of the manuscript that addresses the points raised during the review process.  The study has merit.

Please submit your revised manuscript within 60 days. If you will need more time than this to complete your revisions, please reply to this message or contact the journal office at plosone@plos.org. Please include the following items when submitting your revised manuscript:

We look forward to receiving your revised manuscript.

Kind regards,

Gianfranco D. Alpini

Academic Editor

PLOS ONE

Journal Requirements:

2. Please include additional information regarding the survey or questionnaire used in the study and ensure that you have provided sufficient details that others could replicate the analyses. For instance, if you developed a questionnaire as part of this study and it is not under a copyright more restrictive than CC-BY, please include a copy, in both the original language and English, as Supporting Information. Moreover, please include more details on how the questionnaire was pre-tested, and whether it was validated.

3. In your Methods section, please provide additional information about the participant recruitment method and the demographic details of your participants. Please ensure you have provided sufficient details to replicate the analyses such as: a) the recruitment date range (month and year), b) a description of any inclusion/exclusion criteria that were applied to participant recruitment, c) a table of relevant demographic details, d) a statement as to whether your sample can be considered representative of a larger population, e) a description of how participants were recruited, and f) descriptions of where participants were recruited and where the research took place.Moreover, please report whether sample size calculations were performed.

4.We suggest you thoroughly copyedit your manuscript for language usage, spelling, and grammar. If you do not know anyone who can help you do this, you may wish to consider employing a professional scientific editing service.  

5.We note that you have indicated that data from this study are available upon request. PLOS only allows data to be available upon request if there are legal or ethical restrictions on sharing data publicly. For information on unacceptable data access restrictions, please see http://journals.plos.org/plosone/s/data-availability#loc-unacceptable-data-access-restrictions.

Reviewers' comments:

Reviewer's Responses to Questions

**Comments to the Author**

1. Is the manuscript technically sound, and do the data support the conclusions?

Reviewer #1: Yes

Reviewer #2: Yes

2. Has the statistical analysis been performed appropriately and rigorously? 

Reviewer #1: Yes

Reviewer #2: Yes

3. Have the authors made all data underlying the findings in their manuscript fully available?

Reviewer #1: Yes

Reviewer #2: Yes

4. Is the manuscript presented in an intelligible fashion and written in standard English?

Reviewer #1: No

Reviewer #2: Yes

5. Review Comments to the Author

Reviewer #1: In the current manuscript, Mohammad et al. aimed to study the association between body composition and risk of non-alcoholic fatty liver (NAFLD) development. The authors recruited 2160 patient samples from the nutrition clinics between 2016 and 2017. To determine NAFLD, they used FiroScanTM to determine the controlled attenuation parameter (CAP) score which has been used to define steatosis grade. The body composition was examined by using the dual-energy X-ray absorptiometry (DXA) technique. It was found that fat-free tissue is negatively correlated with NAFLD. Besides, percentages of fat tissue in different body compositions are highly correlated with the risk of NAFLD. Overall, the current study has some clinical importance in guiding the application of fat contents in body composition for the prediction of NAFLD development. However, several issues identified in this manuscript should be further clarified by the authors.

1. The manuscript should be proofread by professional editing service.

2. Table 2 is confused. The authors may consider separating the results into 2 tables including one in males and another one in females.

3. Based on your interpretation, the female would have a high risk of developing NAFLD, if they have total fat and abdominal fat more than 32.23% and 21.42%, respectively. Male would have a high risk of developing NAFLD, if they have total fat and abdominal fat more than 26.73% and 13.76%, respectively. It seems males have a high incidence of developing NAFLD since they require low content of body fat composition to develop NAFLD. In animal studies, it is common that scientists use male mice to study NAFLD because male mice develop NAFLD easily. Could you discuss this aspect in detail in your discussion?

4. Did you check inflammatory cytokines or blood cell contents in your study? It will be interesting to validate if NAFLD patients have a low level of inflammation since adipokines have been linked to inflammation.

Reviewer #2: In this study, Ariya et al. seek the association between body composition with risk of NAFLD. The design and procedures of this study sound reasonable and I have no major comments or criticisms for this manuscript. It would be easier for readers to understand if the authors show ROC curves in figures, not Table 3, with p values. Table 2 is confusing and misleading. It looks like the authors show correlation between male and female (e.g., correlation between male left arm fat and female right arm fat), which does not make sense. I think data Table 1 show combined data of male and female, and the authors want to show data for both genders separately. If so, Table 1 is enough to show without Table 2, and if the authors really need to show male and female data separately, format of tables should be same as Table 1, not combined like current Table 2.

6. PLOS authors have the option to publish the peer review history of their article (what does this mean?). If published, this will include your full peer review and any attached files.

Reviewer #1: No

Reviewer #2: No

---

## [Author Response · Author response to Decision Letter 0]

15 Feb 2021

Response to reviewers 

Dear Editor and Reviewers

We highly appreciate for your detailed and valuable comments on our manuscript. The suggestions were quite helpful for us and we incorporated them in the revised paper. We had referred to literatures and papers and reconstructed the paper to improve the quality of our manuscript. It is important to note that almost all recommendations were corrected and accordingly, we hope these changes could be acceptable to dear editor and respectful reviewers. Our responses to your comments are as follow:

Reviewer 1

1. The manuscript should be proofread by professional editing service.

• The manuscript was reviewed by two experts and we try to correct all grammatical errors. Also certificate for correcting grammatical mistake have sent with manuscript.

2. Table 2 is confused. The authors may consider separating the results into 2 tables including one in males and another one in females.

• Because of the beneficial recommendation of reviewers, table 2 reanalysed with the sex-separated and this analysis was performed in tables 2 and 3.

3. Based on your interpretation, the female would have a high risk of developing NAFLD, if they have total fat and abdominal fat more than 32.23% and 21.42%, respectively. Male would have a high risk of developing NAFLD, if they have total fat and abdominal fat more than 26.73% and 13.76%, respectively. It seems males have a high incidence of developing NAFLD since they require low content of body fat composition to develop NAFLD. In animal studies, it is common that scientists use male mice to study NAFLD because male mice develop NAFLD easily. Could you discuss this aspect in detail in your discussion?

• While thanking the reviewers for pointing out this important point, the following sentence was added to the discussion: "Notably, percentage of total fat in women and men of the same age is remarkably different. In fact, total fat is divided into “essential” and “storage" fats. Essential fat has been defined as the fundamental fat for normal physiologic functioning which consists of around 3% and 12% total body of males and females, respectively. On the other hand, storage fat is considered an energy reserve which is expendable and stored in adipose tissue. Thus, the average total fat of an individual (essential fat plus storage fat) in men and women is estimated to be 18-24 percent and 25-31 percent, respectively. Since the percentage of total fat in women is higher than men due to their physiological condition, the amount of total fat affected in diseases such as NAFLD should typically be higher in them than in males. Similarly, in men, the lower amount of total and abdominal fat affects the development of fatty liver, which does not necessarily mean that they develop NAFLD easily."

4. Did you check inflammatory cytokines or blood cell contents in your study? It will be interesting to validate if NAFLD patients have a low level of inflammation since adipokines have been linked to inflammation.

• We so appreciate for pointing out this point, but in the present study, these tests were not performed due to financial constraints. But this issue was mentioned in the limitations for future studies.

Reviewer 2

In this study, Ariya et al. seek the association between body composition with risk of NAFLD. The design and procedures of this study sound reasonable and I have no major comments or criticisms for this manuscript. It would be easier for readers to understand if the authors show ROC curves in figures, not Table 3, with p values. Table 2 is confusing and misleading. It looks like the authors show correlation between male and female (e.g., correlation between male left arm fat and female right arm fat), which does not make sense. I think data Table 1 show combined data of male and female, and the authors want to show data for both genders separately. If so, Table 1 is enough to show without Table 2, and if the authors really need to show male and female data separately, format of tables should be same as Table 1, not combined like current Table 2.

• We appreciate your valuable comments. Table 3 (now shown as Table 4 in the manuscript) and Figure 1 are reported separately, so it was not possible to display all variables such as "Cut-off Point" in Figure 1. Again, if the reviewer's opinion is to delete this table, we will sincerely delete this table. Besides, it is noteworthy mentioning because of the beneficial recommendation of reviewers, table 2 reanalyzed with the sex-separated and this analysis was performed in tables 2 and 3.

---

## [Decision Letter · Decision Letter 1]

3 Mar 2021

PONE-D-20-34964R1

Assessment of the association between body composition and risk of Non-alcoholic fatty liver

PLOS ONE

Dear Dr. Reza Homayounfar,

Thank you for submitting your manuscript to PLOS ONE. After careful consideration, we feel that it has merit but does not fully meet PLOS ONE’s publication criteria as it currently stands. Therefore, we invite you to submit a revised version of the manuscript that addresses the points raised during the review process. 

Please submit your revised manuscript within 60 days. If you will need more time than this to complete your revisions, please reply to this message or contact the journal office at plosone@plos.org. Please include the following items when submitting your revised manuscript:

We look forward to receiving your revised manuscript.

Kind regards,

Gianfranco D. Alpini

Academic Editor

PLOS ONE

Journal Requirements:

Reviewers' comments:

Reviewer's Responses to Questions

**Comments to the Author**

1. If the authors have adequately addressed your comments raised in a previous round of review and you feel that this manuscript is now acceptable for publication, you may indicate that here to bypass the “Comments to the Author” section, enter your conflict of interest statement in the “Confidential to Editor” section, and submit your "Accept" recommendation.

Reviewer #1: (No Response)

Reviewer #2: All comments have been addressed

2. Is the manuscript technically sound, and do the data support the conclusions?

Reviewer #1: Yes

Reviewer #2: Yes

3. Has the statistical analysis been performed appropriately and rigorously? 

Reviewer #1: Yes

Reviewer #2: Yes

4. Have the authors made all data underlying the findings in their manuscript fully available?

Reviewer #1: Yes

Reviewer #2: Yes

5. Is the manuscript presented in an intelligible fashion and written in standard English?

Reviewer #1: Yes

Reviewer #2: Yes

6. Review Comments to the Author

Reviewer #1: As gender difference has been discussed in NAFLD development, the authors may want to include two tables showing data as the table 1 displayed.

Reviewer #2: In this study, Ariya et al. seek the association between body composition with risk of NAFLD. I have no further comments.

7. PLOS authors have the option to publish the peer review history of their article (what does this mean?). If published, this will include your full peer review and any attached files.

Reviewer #1: No

Reviewer #2: No

---

## [Author Response · Author response to Decision Letter 1]

8 Mar 2021

Response to reviewers 

Dear Editor and Reviewers

On behalf of all the authors of the article, I would like to thank the respected editors and reviewers who considered our article worthy of publication in the prestigious journal of Plos ONE and made suggestions to improve the article. We did our best to correct the recommendations and we hope that the article in its current form will acceptable to dear editor and the respected reviewers.

Comments to the Author

1. If the authors have adequately addressed your comments raised in a previous round of review and you feel that this manuscript is now acceptable for publication, you may indicate that here to bypass the “Comments to the Author” section, enter your conflict of interest statement in the “Confidential to Editor” section, and submit your "Accept" recommendation.

Reviewer #1: (No Response)

Reviewer #2: All comments have been addressed

2. Is the manuscript technically sound, and do the data support the conclusions?

Reviewer #1: Yes

Reviewer #2: Yes

3. Has the statistical analysis been performed appropriately and rigorously?

Reviewer #1: Yes

Reviewer #2: Yes

4. Have the authors made all data underlying the findings in their manuscript fully available?

Reviewer #1: Yes

Reviewer #2: Yes

5. Is the manuscript presented in an intelligible fashion and written in standard English?

Reviewer #1: Yes

Reviewer #2: Yes

6. Review Comments to the Author

Reviewer #1: As gender difference has been discussed in NAFLD development, the authors may want to include two tables showing data as the table 1 displayed.

Reviewer #2: In this study, Ariya et al. seek the association between body composition with risk of NAFLD. I have no further comments.

Considering that changing table number one would cause a significant change in the results section of the article, we decided to make this suggestion of the esteemed reviewer in the form of supplementary tables number one and two.

---

## [Decision Letter · Decision Letter 2]

15 Mar 2021

Assessment of the association between body composition and risk of Non-alcoholic fatty liver

PONE-D-20-34964R2

Dear Dr. Reza Homayounfar,

We’re pleased to inform you that your manuscript has been judged scientifically suitable for publication and will be formally accepted for publication once it meets all outstanding technical requirements.

Kind regards,

Gianfranco D. Alpini

Academic Editor

PLOS ONE

Additional Editor Comments (optional):

Reviewers' comments:

Reviewer's Responses to Questions

**Comments to the Author**

1. If the authors have adequately addressed your comments raised in a previous round of review and you feel that this manuscript is now acceptable for publication, you may indicate that here to bypass the “Comments to the Author” section, enter your conflict of interest statement in the “Confidential to Editor” section, and submit your "Accept" recommendation.

Reviewer #1: All comments have been addressed

Reviewer #2: All comments have been addressed

2. Is the manuscript technically sound, and do the data support the conclusions?

Reviewer #1: Yes

Reviewer #2: Yes

3. Has the statistical analysis been performed appropriately and rigorously? 

Reviewer #1: Yes

Reviewer #2: Yes

4. Have the authors made all data underlying the findings in their manuscript fully available?

Reviewer #1: Yes

Reviewer #2: Yes

5. Is the manuscript presented in an intelligible fashion and written in standard English?

Reviewer #1: Yes

Reviewer #2: Yes

6. Review Comments to the Author

Reviewer #1: Thanks for your response. It is okay to have the table published as supplementary information. It is acceptable.

Reviewer #2: In this study, Ariya et al. seek the association between body composition with risk of NAFLD. I have no further comments.

7. PLOS authors have the option to publish the peer review history of their article (what does this mean?). If published, this will include your full peer review and any attached files.

Reviewer #1: No

Reviewer #2: No

---

## [Editor Report · Acceptance letter]

19 Mar 2021

PONE-D-20-34964R2 

Assessment of the association between body composition and risk of Non-alcoholic fatty liver 

Dear Dr. Homayounfar:

I'm pleased to inform you that your manuscript has been deemed suitable for publication in PLOS ONE. Congratulations! Your manuscript is now with our production department. 

Kind regards, 

on behalf of

Dr. Gianfranco D. Alpini 

Academic Editor

PLOS ONE